# Classification of Chronic Kidney Disease in Sonography Using the GLCM and Artificial Neural Network

**DOI:** 10.3390/diagnostics11050864

**Published:** 2021-05-11

**Authors:** Dong-Hyun Kim, Soo-Young Ye

**Affiliations:** Department of Radiological Science, Graduate School, Catholic University of Pusan, 57 Oryun-daero, Geumjeong-gu, Busan 46252, Korea; dhkim@cup.ac.kr

**Keywords:** kidney ultrasound, gray-level co-occurrence matrix (GLCM), artificial neural network, classification, chronic kidney disease (CKD)

## Abstract

Chronic kidney disease (CKD) can be treated if it is detected early, but as the disease progresses, recovery becomes impossible. Eventually, renal replacement therapy such as transplantation or dialysis is necessary. Ultrasound is a test method with which to diagnose kidney cancer, inflammatory disease, nodular disease, chronic kidney disease, etc. It is used to determine the degree of inflammation using information such as the kidney size and internal echo characteristics. The degree of the progression of chronic kidney disease in the current clinical trial is based on the value of the glomerular filtration rate. However, changes in the degree of inflammation and disease can even be observed with ultrasound. In this study, from a total of 741 images, 251 normal kidney images, 328 mild and moderate CKD images, and 162 severe CKD images were tested. In order to diagnose CKD in clinical practice, three ROIs were set: the cortex of the kidney, the boundary between the cortex and medulla, and the medulla, which are areas examined to obtain information from ultrasound images. Parameters were extracted from each ROI using the GLCM algorithm, which is widely used in ultrasound image analysis. When each parameter was extracted from the three areas, a total of 57 GLCM parameters were extracted. Finally, a total of 58 parameters were used by adding information on the size of the kidney, which is important for the diagnosis of chronic kidney disease. The artificial neural network (ANN) was composed of 58 input parameters, 10 hidden layers, and 3 output layers (normal, mild and moderate CKD, and severe CKD). Using the ANN model, the final classification rate was 95.4%, the epoch needed for training was 38 times, and the misclassification rate was 4.6%.

## 1. Introduction

Chronic kidney disease (CKD) is increasing with the aging of the population and the increase in chronic diseases. In many countries, it is a concerning health problem causing a high prevalence and incidence of stroke, heart disease, and complications such as diabetes and infection, as well as increases in medical expenses [1,2]. Chronic kidney disease is used as a general term that refers to kidney damage for more than 3 months or a continuous decrease in kidney function regardless of the cause. It is a disease that increases the risk of cardiocerebrovascular disease and is accompanied by kidney failure and various complications [3,4]. The following definition of chronic kidney disease is widely used by the National Kidney Foundation (NKF): “if the kidney damage, such as proteinuria, hematuria, or pathological abnormality, or glomerular filtration rate is less than 60 mL/min/1.73 m^2^ and lasts for more than 3 months” [5]. Chronic kidney disease has various causes and pathological findings, but clinically, it can be confirmed relatively simply through a urine test to detect proteinuria and a blood test to estimate the glomerular filtration rate. In the clinical field, the stage of chronic kidney disease is divided into five stages from stage 1 to stage 5 according to the estimated glomerular filtration rate for diagnosis and treatment of patients. Chronic kidney disease can be said to be a relatively common chronic disease, but compared with other chronic diseases such as high blood pressure and diabetes, the general public’s awareness is not high [6]. Because it can be easily diagnosed through a relatively simple test, early diagnosis can slow the progression of the disease through early treatment. However, because the symptoms are not clear, the disease is often diagnosed after a long period of time, leading to kidney failure or even death due to complications such as cardiocerebrovascular disease. In general, chronic kidney disease continues to progress once it occurs. However, early diagnosis can slow the decline in kidney function and recover from chronic kidney disease [7].

Ultrasound, which is one of the methods for diagnosing chronic kidney disease, is very important for checking if the kidney function is decreased due to other reasons, such as the size and morphology of the kidney [8].

The GLCM algorithm, which extracts feature points by emphasizing spatial characteristics in such ultrasound images, is widely used in ultrasound image analysis by using the correlation between the current pixel and the brightness value of its neighboring pixels [9].

The automatic detection of diseases using images such as ultrasound graphics via computational methods has been of great importance in recent years, especially after the creation of computer-aided diagnosis systems (CADs) [10,11].

Artificial neural networks (ANNs) are usually employed to formulate statistical models for computer analysis. An artificial neural network (ANN) or a simple neural network (NN) is a computational structure based on the functioning of biological neurons. Networks of this type have the ability to learn and recognize patterns based on input information and use these patterns to predict future information by extrapolating the learned data. These neural networks consist of small processing units called neurons connected with one another in groups called layers; the common neural network has at least one input layer, one hidden layer, and one output layer [12].

In this paper, various parameters were obtained by applying the GLCM algorithm to ultrasound images classified as normal kidney images, mild and moderate CKD images, and severe CKD images. Chronic kidney disease was classified by applying an artificial neural network using GLCM parameters and kidney size. The purpose of this study is to classify chronic kidney disease into three categories to determine whether early diagnosis of chronic kidney disease is possible.

## 2. Materials and Methods

### 2.1. Subjects and Data Acquisition

From January 2015 to December 2017, the experiment was performed using ultrasound images of volunteers who visited the hospital for medical examination at R Hospital in Yangsan, Gyeong-sang-nam-do, and patients diagnosed with CKD.

Table 1 shows the clinical characteristics of subjects. A total of 741 ultrasound images were used, including 251 normal images, 328 mild and moderate CKD images, and 162 severe CKD images, classified by the medical doctor. Radiologists classified the experimental images into those of mild and moderate CKD patients and severe CKD patients by reading, and cases with kidney cancer, renal cysts, and congenital diseases of the kidney were excluded from the experiment [13].

Regarding the experimental equipment, an RS80A (Samsung Medison, Seoul, Korea) ultrasound device was used, and the image conditions were acquired with a gain of 50, dynamic range of 56, frame average of 8, power of 95, and general frequency. All tests were conducted under voluntary participation and were approved by the Institutional Review Board (IRB) of the Catholic University of Busan (CUPIRB-2017-023).

Table 1 shows the information on the age, sex, eGFR, hypertension, and diabetes of the test subjects.

### 2.2. Experimental Method

The region of interest (ROI) was set in the ultrasound image of the kidney, and the result was calculated using MATLAB 2016a (MathWorks Inc., Natick, MA, USA). The ROI was selected by setting the cortical region, the boundary region between the cortex and medulla, and the medulla region. In the ultrasound image, the characteristics of the normal kidney are in the shape of an oval composed of the renal cortex, which shows low echo compared to the liver, and the renal sinus, which shows high echo. The boundary between the cortex and the renal sinus is clear, and a high echo in the center is visible. On the other hand, in the kidney that is continuously damaged, the ultrasonic echo of the renal cortex increases due to fibrosis. As a result, the boundary between the brightened renal cortex and the renal cortex is unclearly observed. In addition, due to the decline in function, the size decreases, and kidney atrophy is observed [14]. Because these features are used in the diagnosis of chronic kidney disease, the ROI area was set to 50 × 50 and set to 3 locations. The following figure shows the normal, moderate, and severe ultrasound images of 872 × 1280 resolution used in this study. Figure 1 shows normal, mild and moderate and severe CKD ultrasound images.

Figure 2 shows the overall block diagram of this experiment.

Histogram equalization and range filter preprocessing were performed on the original ultrasound image. Histogram equalization can improve the quality of an image by making the distribution of light and dark in the image uniform [15]. The range filter can cause the noise reduction effect by highlighting the boundary of the image [16]. After preprocessing was performed, feature parameters were extracted by applying the GLCM algorithm. A total of 19 parameters were extracted from each of the three regions of the cortex, the boundary between the cortex and medulla, and the medulla, and, finally, the size information was combined to determine a total of 58 input parameters.

An artificial neural network with 10 hidden layers was constructed, and the output was designed to be classified into three types: normal, mild and moderate CKD, and severe CKD.

### 2.3. Gray-Level Co-Occurrence Matrix (GLCM)

The GLCM [17], a texture descriptor, is used to compute the second-order statistical features from normal and CKD images. Considering an image I with a size of M × N and Ng number of distinct gray levels, the variations of texture are calculated by using the gray tone spatial dependence matrix *p* (*i, j*), where the pixels are separated with a distance d at the *i*-th and *j*-th gray levels. In this present work, four angles (0°, 45°, 90°, and 135°) with a pixel distance of 1 were considered. The second-order statistical features (autocorrelation, contrast, correlation, cluster prominence, cluster shade, dissimilarity, energy, entropy, homogeneity, maximum probability, sum of squares variance, sum average, sum variance, sum entropy, difference variance, difference entropy, information measure of correlation 1, information measure of correlation 2, and inverse difference (INV)) were calculated using the GLCM. Figure 3a shows the result implemented using MATLAB to obtain GLCM parameters. If you select the ROI area and run, it is saved as an Excel file, as shown in Figure 3b.

Table 3 shows the parameters that can be extracted when calculating the GLCM using an ROI image. Table 1 shows the variables and notation used to compute texture features that are the parameters of the GLCM.

We used the equations in Table 3 that are based on second-order statistical values, obtaining 19 values for each image, corresponding to 19 different texture descriptors. These calculations were performed using the operations indicated in Table 1 for each position *p* (*i*, *j*) of the GLCM and adding all of the values from the GLCM. Prior to obtaining these features, a few simple calculations are needed. $\mu_{i}$, $\mu_{j}$ and $\sigma_{i}$, $\sigma_{j}$ correspond to the mean and standard deviation, respectively, of *i* and *j*. $p_{x}\left(i\right)$ is the marginal probability of *i* in $p_{x}$, and $p_{y}\left(j\right)$ is the marginal probability of *j* in $p_{y}$. *HX* is the entropy of $p_{x}$, and *HY* is the entropy of $p_{y}$. These values were calculated using Table 2 [18].

### 2.4. Artificial Neural Network (ANN)

The ANN is an element of machine learning that has currently become significant in research and development. The concept of machine learning is the ability of a computer to understand the structure of data using a mathematical or statistical model. The foundation of an ANN is made up of a single layer of input, process, and output elements. As a result, from a very basic concept of an information processing cycle, ANN then performs a complex mathematical formulation in order to produce an optimum result for any dataset or problem segment [19].

Figure 4 shows the block diagram of the neural network used in this experiment. ROI was selected from 3 locations for feature detection in the original image.

ROI was selected from the cortex, the boundary between the cortex and medulla, and the medulla. The size of the kidney, which is a very important factor in the diagnosis of chronic kidney disease, was selected as a feature parameter. As a result of the GLCM calculation, 19 parameters were extracted from each of the cortex regions; that is, the boundary between the cortex and medulla and the medulla (Table 3). In addition, by adding the size of the kidney as a parameter, the input layer consisted of a total of 58 nodes. After passing through 10 hidden layers, the output layer was classified into three types: normal, mild and moderate CKD, and severe CKD.

## 3. Results

The implementation of image preprocessing, the GLCM algorithm, and the artificial neural network was carried out using MATLAB R2016a. This tool provides a user-friendly interface and has many inbuilt functions, so it is easy to implement algorithms in it. Windows 10 (64 bit) with a 3.60 GHz Intel i9 processor and 64 GB of RAM was used for this study.

Figure 5 shows the result of setting three ROIs in the original image.

Figure 6 shows the results of applying histogram equalization to the ROI area during the pretreatment process.

After the preprocessing of US images, texture statistical features were extracted using the GLCM algorithm. Means of features were obtained using the GLCM relative to four different orientations that were calculated. Table 4, Table 5, Table 6 and Table 7 display the GLCM results of the cortex, the boundary between the cortex and medulla, and the medulla, respectively. Table 7 shows the size of the kidney that is normal and that with mild and moderate CKD and severe CKD.

Figure 7 shows the result of the ANN consisting of 64 inputs, 10 hidden layers, and 3 outputs. In the Figure 7a, the training result was 95.6%, the validation result was 97.3%, and the test result was 85.7%. The implemented modeling has an error of 0.030511 as shown in Figure 7b. Using this ANN model, the final classification rate was 95.4%. Figure 8 shows the ROC curve for the results.

## 4. Discussion

Worldwide, the number of patients with chronic kidney disease is increasing at a tremendous rate. Chronic kidney disease is particularly commonly observed in conjunction with diabetes, high blood pressure, and old age, but Korea has recently become an aging society, and the number of patients with high blood pressure and diabetes has increased due to the Westernized lifestyle; additionally, about 10% of the adult population suffers from chronic kidney disease and is 60 years old or older. The incidence of chronic kidney disease is increasing rapidly. This phenomenon means that the number of patients who need dialysis or a kidney transplant due to chronic kidney disease is also increased, and when the number of patients with end-stage renal failure increases, a huge loss is inevitable in terms of both the social and national contexts as well as in individual patients and families [2].

Looking at the studies for diagnosing chronic kidney disease, the 2017 Igbinedion [20] study and the 2020 Prashanth [21] study determined the CKD stage using creatinine levels and eGFR, which are common diagnostic criteria in clinical practice. In 2013, Dijana et al. [8] diagnosed kidney disease by measuring kidney size according to kidney function and anthropometric characteristics. There was a significant correlation between all measured kidney dimensions, volume, parenchymal thickness, and serum creatinine. In 2020, Priyanke et al. [22] conducted a study to extract feature parameters from the kidney by applying the GLCM and PCA to ultrasound images. This study focuses on the preprocessing process and how feature parameters can be extracted from the kidney. In 2019, Kuo et al. [23] used the transfer learning technique, integrating the powerful ResNet model pretrained on an ImageNet dataset in neural network architecture, to predict kidney function based on 4505 kidney ultrasound images labeled using eGFRs derived from serum creatinine concentrations. Since the mid–late 2000s, many studies have started to apply machine learning and deep learning to medical images for diagnosis.

In this study, the cortex of the kidney, the boundary between the cortex and medulla, and the medulla were set as ROIs for diagnosing chronic kidney disease on ultrasound images. Parameters were extracted from each ROI region using the GLCM algorithm, which is widely used in ultrasound image analysis. The parameters are autocorrelation, contrast, correlation, cluster prominence, cluster shade, dissimilarity, energy, entropy, homogeneity, maximum probability, sum of squares variance, sum average, sum variance, sum entropy, difference variance, difference entropy, information measure of correlation 1, information measure of correlation 2, and inverse difference (INV). When each parameter was extracted from three areas, a total of 57 GLCM parameters were extracted. Finally, a total of 58 parameters were used by adding information on the size of the kidney, which is important for the diagnosis of chronic kidney disease. A total of 58 input parameters were tested by constructing an ANN, which is a machine learning method. Input parameters were set to 58, and the hidden layer was set to 10. Because 10 or more hidden layers showed no effect on the classification rate, the experiment was conducted with 10 hidden layers. The three outputs to be classified were normal, mild and moderate CKD, and severe CKD. It is thought that the classification rate of the result to be classified was increased by using all 58 input parameters. The classification accuracy was 95.4%, the epoch needed for training was 38 times, and the misclassification rate was 4.6%. In this experiment, three types of conditions were classified, namely, normal, mild and moderate CKD, and severe CKD, but a further detailed classification of disease states is required. In addition, the experiment was conducted by acquiring 741 data items, which is due to the fact that the amount of data is not large; thus, the machine learning method was selected. When a large amount of data is acquired, we plan to apply the deep learning method after acquiring more data. In order to actually use it clinically, it is necessary to diversify the types of classification. After making the implemented result available to the ultrasonic equipment, it needs to be upgraded through feedback from the sonographer.

## 5. Conclusions

Chronic kidney disease can be treated if it is detected early, but as the disease progresses, recovery becomes impossible. Eventually, renal replacement therapy such as transplantation or dialysis must be used. In other words, it is crucial to detect and treat chronic kidney disease in the early stages. Ultrasound is a test method for diagnosing kidney cancer, inflammatory disease, nodular disease, chronic kidney disease, etc., and is used to check information on the degree of inflammation using information such as kidney size and internal echo characteristics.

In this study, ultrasound images, including 251 normal kidney images, 328 mild and moderate kidney disease images, and 162 severe renal kidney images, were used in 741 cases. In order to diagnose chronic kidney disease in clinical practice, three ROIs were set, namely, the cortex of the kidney, the boundary between the cortex and medulla, and the medulla, which are areas examined to obtain information from ultrasound images. Parameters were extracted from each ROI using the GLCM algorithm, which is widely used in ultrasound image analysis. When each parameter was extracted from the three areas, a total of 57 GLCM parameters were extracted. Finally, a total of 58 parameters were used by adding information on the size of the kidney, which is important for the diagnosis of chronic kidney disease. The ANN was composed of 58 input parameters, 10 hidden layers, and 3 output layers (normal, mild and moderate CKD, and severe CKD). Using the ANN model, the final classification rate was 95.4%, the epoch needed for training was 38 times, and the misclassification rate was 4.6%. It is believed that this experiment can be used as a basis for implementing an automatic diagnosis system in the area of diagnosing chronic kidney disease using ultrasound images. In addition, the use of experimental results is thought to play an important role in clinical decision making, including early diagnosis and treatment of chronic kidney disease.

## Figures and Tables

**Figure 1 diagnostics-11-00864-f001:**
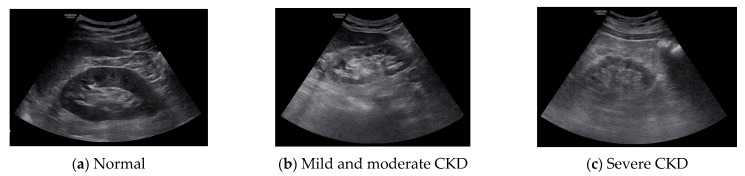
Original images.

**Figure 2 diagnostics-11-00864-f002:**
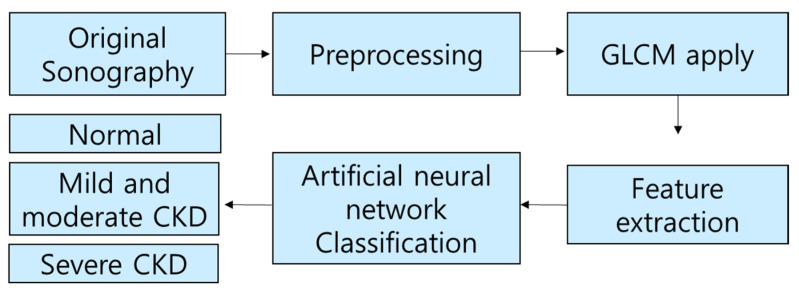
Total block diagram of image processing.

**Figure 3 diagnostics-11-00864-f003:**
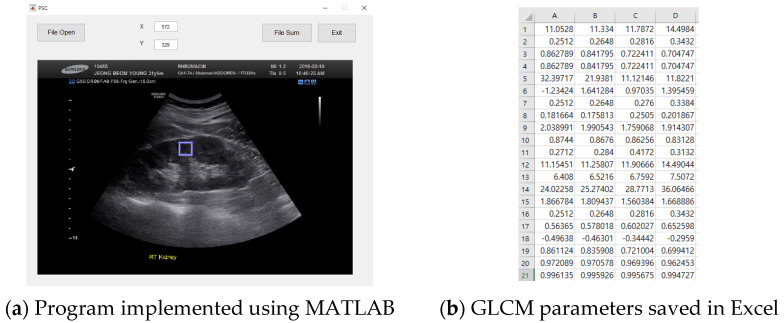
Program implementation.

**Figure 4 diagnostics-11-00864-f004:**
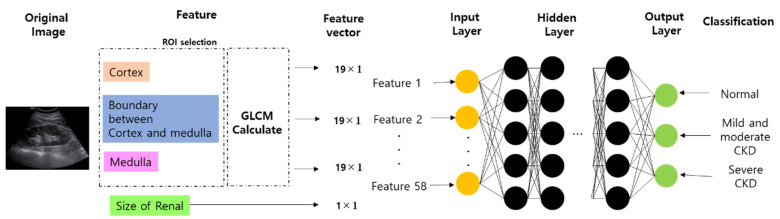
Output classification based on the ANN model.

**Figure 5 diagnostics-11-00864-f005:**
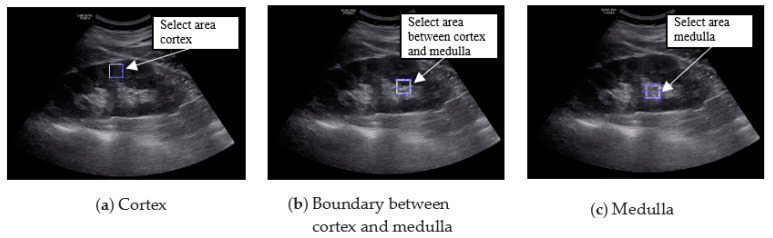
Selection of ROI.

**Figure 6 diagnostics-11-00864-f006:**
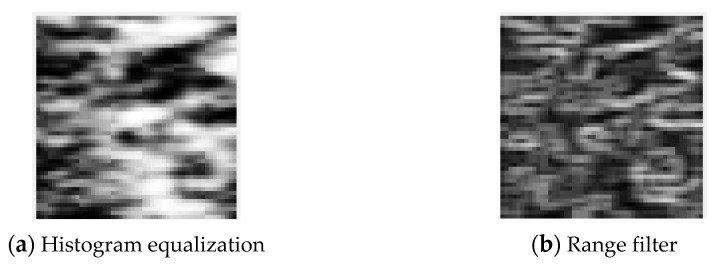
The results of preprocessing: (**a**) histogram equalization; (**b**) range filter.

**Figure 7 diagnostics-11-00864-f007:**
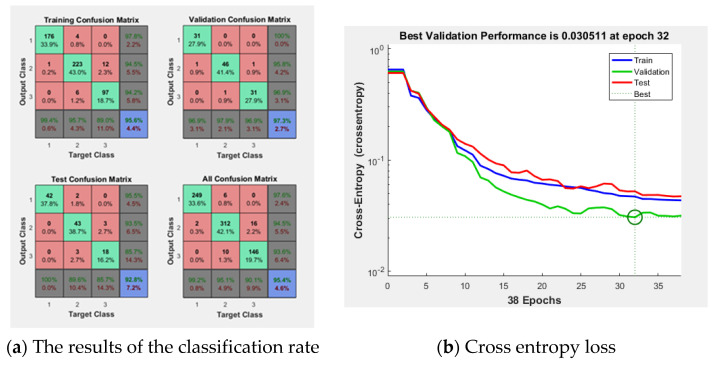
The results of the ANN.

**Figure 8 diagnostics-11-00864-f008:**
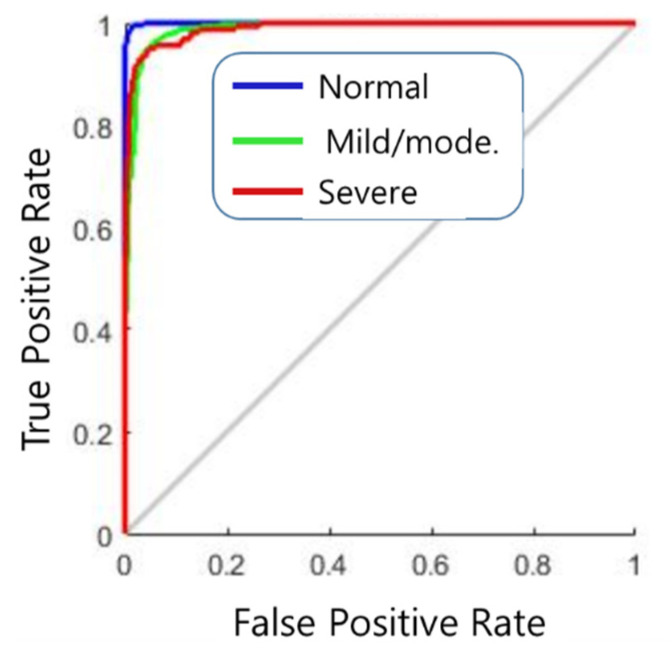
ROC curve of the results.

**Table 1 diagnostics-11-00864-t001:** Clinical characteristics.

Class (Number)	Range of Age	Number	Gender	Number	eGFR	Hyper. (Number)	Diabetes (Number)
Normal (251)	30–45	82	F	102	105 ± 16.7	74	82
	55–65	107					
			M	149			
	65–70	62					
Mild & Moderate (328)	45–55	43	F	125	40 ± 9.7	328	328
	55–65	179					
			M	203			
	65–70	106					
Severe (162)	55–60	95	F	67	4 ± 4.2	162	162
	60–70	67	M	95			

**Table 2 diagnostics-11-00864-t002:** Variables and notations used to compute the texture features.

Notation	Definition
*x* (*i*, *j*)	Element *i*, *j* in the GLCM
N	Number of gray levels
*p* (*i*, *j*)	$\frac{x\left(i,j\right)}{\sum_{i=1}^{N}\sum_{j=1}^{N}x\left(i,j\right)}$
$p_{x}\left(i\right)$	$\sum_{j=1}^{N}p\left(i,j\right)$
$p_{y}\left(j\right)$	$\sum_{i=1}^{N}p\left(i,j\right)$
$\mu_{x}$	$\sum_{i=1}^{N}i\cdot p_{x}\left(i\right)$
$\mu_{y}$	$\sum_{j=1}^{N}j\cdot p_{y}\left(j\right)$
$\sigma_{x}^{2}$	$\sum_{i=1}^{N}{\left(i-\mu_{x}\right)}^{2}\cdot p_{x}\left(i\right)$
$\sigma_{y}^{2}$	$\sum_{j=1}^{N}{\left(j-\mu_{y}\right)}^{2}\cdot p_{y}\left(i\right)$
$p_{x+y}\left(k\right)$	$\sum_{i=}^{N}\sum_{j=1\;}^{N}{\;}_{k=i+j}\;p\left(i,j\right)$
$p_{x-y}\left(k\right)$	$\sum_{i=}^{N}\sum_{j=1\;}^{N}{\;}_{k=\left|i-j\right|}\;p\left(i,j\right)$
$\mu_{x+y}$	$\sum_{k=2}^{2N}k\cdot p_{x+y}\left(k\right)$
$\mu_{x-y}$	$\sum_{k=0}^{N-1}k\cdot p_{x-y}\left(k\right)$
HX	$-\sum_{i=1}^{N}p_{x}\left(i\right)\cdot \log p_{x}\left(i\right)$
HY	$-\sum_{i=1}^{N}p_{y}\left(i\right)\cdot \log p_{y}\left(i\right)$
HXY	$-\sum_{i=1}^{N}\sum_{j=1}^{N}p\left(i,j\right)\cdot \log p\left(i,j\right)$
HXY1	$-\sum_{i=1}^{N}\sum_{j=1}^{N}p\left(i,j\right)\cdot \log\left[p_{x}\left(i\right)\cdot p_{y}\left(j\right)\right]$
HXY2	$-\sum_{i=1}^{N}\sum_{j=1}^{N}p_{x}\left(i\right)\cdot p_{y}\left(j\right)\cdot \log\left[p_{x}\left(i\right)\cdot p_{y}\left(j\right)\right]$

**Table 3 diagnostics-11-00864-t003:** Parameters of the GLCM.

Parameter	Formula
Autocorrelation	$\sum_{i=1}^{N}\sum_{j=1}^{N}\left(i\right)\left(j\right)p\left(i,j\right)$
Contrast	$\sum_{i=1}^{N}\sum_{j=1}^{N}{\left(i-j\right)}^{2}p\left(i,j\right)$
Correlation	$\sum_{i=1}^{N}\sum_{j=1}^{N}\frac{\left(i-\mu_{x}\right)\left(j-\mu_{y}\right)p\left(i,j\right)}{\sigma_{x}\sigma_{y}}$
Cluster prominence	$\sum_{i=1}^{N}\sum_{j=1}^{N}{\left(i+j-2\mu \right)}^{3}p\left(i,j\right)$
Cluster shade	$\sum_{i=1}^{N}\sum_{j=1}^{N}{\left(i+j-2\mu \right)}^{4}p\left(i,j\right)$
Dissimilarity	$\sum_{i=1}^{N}\sum_{j=1}^{N}\left|i-j\right|p\left(i,j\right)$
Energy	$\sum_{i=1}^{N}\sum_{j=1}^{N}{\left(p\left(i,j\right)\right)}^{2}$
Entropy	$-\sum_{i=1}^{N}\sum_{j=1}^{N}p\left(i,j\right)\log\left(p\left(i,j\right)\right)$
Homogeneity	$-\sum_{i=1}^{N}\sum_{j=1}^{N}\frac{p\left(i,j\right)}{1+{\left(i-j\right)}^{2}}$
Maximum probability	$\max_{\left(ij\right)}p\left(i,j\right)$
Sum of squares	$\sum_{i=1}^{N}\sum_{j=1}^{N}{\left(i-\mu \right)}^{2}p\left(i,j\right)$
Sum average	$\sum_{i=2}^{2N}iP_{x+y}\left(i\right)$
Sum variance	$\sum_{i=2}^{2N}{\left(i-\mu_{x+y}\right)}^{2}P_{x+y}\left(i\right)$
Sum entropy	$-\sum_{i=2}^{2N}P_{x+y}\left(i\right)\log p_{x+y}\left(i\right)$
Difference variance	$\sum_{i=1}^{N}{\left(k-\mu_{x-y}\right)}^{2}p_{x-y}\left(i\right)$
Difference entropy	$-\sum_{i=1}^{N}p_{x-y}\left(i\right)\log\left(p_{x-y}\left(i\right)\right)$
Information measure of correlation 1	$\frac{HXY-HXY1}{\max\left(HX,HY\right)}$
Information measure of correlation 2	$\sqrt{1-\exp\left[-2\left(HXY2-HXY\right)\right]}$
Inverse difference (INV)	$\sum_{i=1}^{N}\sum_{j=2}^{N}\frac{p\left(i,j\right)}{1+\left|i-j\right|}$

**Table 4 diagnostics-11-00864-t004:** Values of GLCM features of the cortex (mean ± STD).

Variable	Normal	Mild & Mod. CKD	Severe CKD
Autocorrelation	2.40 ± 1.06	6.13 ± 2.88	7.59 ± 2.73
Contrast	0.15 ± 0.07	0.17 ± 0.06	0.18 ± 0.04
Correlation	0.62 ± 0.14	0.66 ± 0.13	0.77 ± 0.07
Cluster prominence	4.55 ± 13.16	5.75 ± 9.19	9.08 ± 7.12
Cluster shade	0.81 ± 1.90	0.75 ± 1.46	1.31 ± 1.13
Dissimilarity	0.15 ± 0.07	0.17 ± 0.06	0.18 ± 0.04
Energy	0.52 ± 0.20	0.44 ± 0.16	0.35 ± 0.10
Entropy	0.99 ± 0.39	1.21 ± 0.35	1.44 ± 0.26
Homogeneity	0.92 ± 0.036	0.91 ± 0.03	0.90 ± 0.02
Maximum probability	2.43 ± 1.07	6.16 ± 2.78	7.58 ± 2.71
Sum of squares variance	2.92 ± 0.66	4.73 ± 1.14	5.30 ± 0.94
Sum average	5.18 ± 2.53	15.35 ± 7.78	18.10 ± 7.39
Sum variance	0.89 ± 0.34	1.09 ± 0.31	1.32 ± 0.24
Sum entropy	0.15 ± 0.07	0.17 ± 0.06	0.18 ± 0.04
Difference variance	0.40 ± 0.144	0.45 ± 0.10	0.47 ± 0.07
Difference entropy	−0.32 ± 0.13	−0.35 ± 0.10	−0.43 ± 0.07
Information measure of correlation 1	0.54 ± 0.15	0.62 ± 0.14	0.73 ± 0.08
Information measure of correlation 2	0.98 ± 0.008	0.98 ± 0.006	0.98 ± 0.005
Inverse difference (INV)	1.00 ± 0.001	1.00 ± 0.0009	1.00 ± 0.0007

**Table 5 diagnostics-11-00864-t005:** Values of GLCM features of the boundary between the cortex and medulla (mean ± STD).

Variable	Normal	Mild & Mod. CKD	Severe CKD
Autocorrelation	6.60 ± 2.43	9.05 ± 3.13	10.36 ± 3.78
Contrast	0.20 ± 0.07	0.20 ± 0.05	0.21 ± 0.05
Correlation	0.88 ± 0.05	0.82 ± 0.09	0.75 ± 0.10
Cluster prominence	45.43 ± 40.19	21.83 ± 24.01	9.42 ± 8.02
Cluster shade	3.88 ± 4.37	2.11 ± 3.35	0.86 ± 1.23
Dissimilarity	0.20 ± 0.05	0.20 ± 0.05	0.21 ± 0.05
Energy	0.25 ± 0.11	0.28 ± 0.09	0.31 ± 0.07
tropy	1.80 ± 0.28	1.66 ± 0.29	1.54 ± 0.25
Homogeneity	0.90 ± 0.03	0.90 ± 0.02	0.90 ± 0.02
Maximum probability	6.65 ± 2.42	9.06 ± 3.12	10.35 ± 3.80
Sum of squares variance	4.70 ± 0.92	5.72 ± 1.07	6.22 ± 1.16
Sum average	13.58 ± 5.86	21.20 ± 8.48	26.00 ± 10.77
Sum variance	1.66 ± 0.26	1.52 ± 0.27	1.40 ± 0.23
Sum entropy	0.20 ± 0.05	0.20 ± 0.05	0.21 ± 0.05
Difference variance	0.48 ± 0.10	0.49 ± 0.07	0.51 ± 0.07
Difference entropy	−0.55 ± 0.12	−0.49 ± 0.10	−0.41 ± 0.10
Information measure of correlation 1	0.86 ± 0.06	0.80 ± 0.09	0.73 ± 0.09
Information measure of correlation 2	0.98 ± 0.005	1.00 ± 0.006	0.98 ± 0.005
Inverse difference (INV)	1.00 ± 0.0008	1.00 ± 0.0008	1.00 ± 0.0008

**Table 6 diagnostics-11-00864-t006:** Values of GLCM features of the medulla (mean ± STD).

Variable	Normal	Mild & Mod. CKD	Severe CKD
Autocorrelation	11.76 ± 3.39	12.17 ± 3.14	13.23 ± 4.48
Contrast	0.28 ± 0.05	0.22 ± 0.04	0.23 ± 0.05
Correlation	0.74 ± 0.09	0.70 ± 0.11	0.71 ± 0.09
Cluster prominence	17.34 ± 16.03	6.56 ± 4.63	8.38 ± 8.90
Cluster shade	0.69 ± 1.39	0.21 ± 0.90	0.38 ± 1.31
Dissimilarity	0.27 ± 0.045	0.22 ± 0.04	0.23 ± 0.04
Energy	0.25 ± 0.083	0.32 ± 0.08	0.32 ± 0.09
Entropy	1.80 ± 0.32	1.51 ± 0.20	1.53 ± 0.26
Homogeneity	0.86 ± 0.02	1.00 ± 0.02	0.89 ± 0.02
Maximum Probability	11.77 ± 3.39	12.18 ± 3.15	13.22 ± 4.47
Sum of squares variance	6.64 ± 0.99	6.83 ± 0.90	7.09 ± 1.18
Sum average	28.46 ± 9.59	32.15 ± 9.16	35.27 ± 13.52
Sum variance	1.60 ± 0.29	1.35 ± 0.19	1.38 ± 0.23
Sum entropy	0.28 ± 0.05	0.22 ± 0.04	0.23 ± 0.05
Difference variance	0.59 ± 0.06	0.52 ± 0.05	0.53 ± 0.06
Difference entropy	−0.37 ± 0.07	−0.35 ± 0.09	−0.35 ± 0.08
Information measure of correlation 1	0.73 ± 0.10	0.68 ± 0.11	0.69 ± 0.09
Information measure of correlation 2	0.97 ± 0.005	0.98 ± 0.004	0.97 ± 0.005
Inverse difference (INV)	1.00 ± 0.0007	1.00 ± 0.0006	1.00 ± 0.0007

**Table 7 diagnostics-11-00864-t007:** The size of the normal kidney and that with mild and moderate CKD and severe CKD (mean ± STD).

Variable	Normal	Mild & Mod. CKD	Severe CKD
Size	11.7 ± 0.46	9.14 ± 2.00	7.07 ± 1.70

## Data Availability

The data are not publicly available due to privacy.

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
