# Peer review of "Classification of Chronic Kidney Disease in Sonography Using the GLCM and Artificial Neural Network"

_diagnostics, 2021, doi:10.3390/diagnostics11050864_

Round 1

Reviewer 1 Report

General Comments

This is a mono-centre experimental study that tests a new classification of chronic kidney disease in sonography using the GLCM and artificial neural network.

Specific revision comments

  • The introduction and discussion section are misleading and are not focused on the topic.
  • In Introduction (page 2, line 72-75) this statement is out of context, it should be erased or moved in the methods.
  • In Introduction, (page 2 line 75-77), the aim of study is unclear, what is the meaning of "to estimate the degree of CKD"? Please, re-edited this statement.
  • In Methods, (page 2, line 82-83), the terms "early" and "end-stage" CKD are too generic. How did Authors identify the different stages of CKD? Authors should better define this point in the text.
  • In Methods, what were the inclusion and exclusion criteria of study?
  • In Methods, what are the clinical characteristics (ie age, sex, comorbidity, cause of CKD etc) and laboratory tests (GFR, creatinine, proteinuria, etc) of patients and volunteers? A table should be performed
  • In Discussion, (page 9, line 219 -246) this paragraph is redundant because Authors have already described this point in Introduction. It should be re-edited and moved in Introduction. On the beginning of discussion, Authors should summarize the novelty of their research.
  • In Discussion, (page 9 line 247-266) this paragraph is limited to the description of other studies. Authors should compare their results with others, putting in evidence the differences and the value of their data. It should be re-edited
  • I suggest English editing to improve the readability of the manuscript.

Author Response

Response to Reviewer 1 Comments

Thank you for your delicate and caring dissertation review.

I will do the best I can to answer.

Point 1: The introduction and discussion section are misleading and are not focused on the topic.

Response 1:

The introduction and discussion are being revised through the MDPI English service.

We will change the thesis as soon as it is revised.

Please wait for the English revised version to arrive.

Point 2: In Introduction (page 2, line 72-75) this statement is out of context, it should be erased or moved in the methods.

Response 2:

Perhaps this is what the author is referring to.

“In this paper, various parameters were obtained by applying the GLCM algorithm to ultrasound images classified as normal kidney images, early CKD images, and terminal CKD images. Chronic kidney disease was classified by applying artificial neural network using GLCM parameters and kidney size. The purpose of this study was to estimate the degree of chronic kidney disease to determine whether or not chronic kidney disease can be diagnosed at an early stage.”

At the end of many papers, a brief summary of the papers is provided. Like reference 2, 6, 7, 8, 9, 10, 11 etc. These formats are very common to summarize the purpose and method of this paper at the end of the introduction.

If there is a problem with the English content, I will fix it.

Point 3: In Introduction, (page 2 line 75-77), the aim of study is unclear, what is the meaning of "to estimate the degree of CKD"? Please, re-edited this statement.

Response 3:

Perhaps this is what the author is referring to.

“The purpose of this study was to estimate the degree of chronic kidney disease to determine whether or not chronic kidney disease can be diagnosed at an early stage.”

We re-edited as followed.

“The purpose of this study is to classify chronic kidney disease into three categories to determine whether early diagnosis of chronic kidney disease is possible.”

Point 4: In Methods, (page 2, line 82-83), the terms "early" and "end-stage" CKD are too generic. How did Authors identify the different stages of CKD? Authors should better define this point in the text.

Response 4:

Renamed as follows, referring to the contents of the CKD classification method mentioned in reference 2 paper.

Reference 2 “The contribution of chronic kidney disease to the global burden of major nocommunicable disease”

“CKD is classified into stages 1–5, with stages 1 and 2 requiring the presence of kidney damage such as proteinuria as well as reduced GFR.8 Many authors now refer to ‘moderate,’ or clinically significant, CKD as stages 3 (GFR 30–59 ml/min) and 4 (GFR 15–29 ml/min), with o60 ml/min chosen as a cut off because it represents loss of about 50% of normal renal function, although there is now ample evidence of increased risk in earlier stages.9,10

The table below shows a table that categorizes CKD.

5 Stages of Kidney Disease

Kidney Function/GFR

Description

Stage 1

> 90%

Normal or High Function

Stage 2

60-89%

Mildly Decreased Function

Stage 3

30-59%

Mild to Moderately

Decreased Function

Stage 4

15-29%

Severely Decreased

Function

Stage 5

<15%

Kidney Failure

The names re-named are as follows. “Mild and Moderate, Severe”

All changes were made to this classification name in the paper.

Point 5: In Methods, what were the inclusion and exclusion criteria of study?

Response 5:

As the eGFR value changes, the ultrasound image shows a change. There are many papers that have been researched using these features.

References 14 referenced in this paper have been added.

The contents referenced in Reference 14 are as follows.

“Investigators at MSKCC tracked nephrectomy use in 1533 patients between 2000 and 2007 excluding patients with bilateral tumors and tumors in a solitary kidney and including only patients with an eGFR of greater than 45 ml/min/1.73 m2.”

Point 6: In Methods, what are the clinical characteristics (ie age, sex, comorbidity, cause of CKD etc) and laboratory tests (GFR, creatinine, proteinuria, etc) of patients and volunteers? A table should be performed

Response 6:

Table 1 added. Hypertension and diabetes influencing CKD were indicated.

Point 7: In Discussion, (page 9, line 219 -246) this paragraph is redundant because Authors have already described this point in Introduction. It should be re-edited and moved in Introduction.

Response 7:

The same content mentioned in the introduction has been deleted.

Point 8: On the beginning of discussion, Authors should summarize the novelty of their research.

Response 8:

Discussion is the part that explains the subject of one's research presented in the Introduction.

In other words, in the paper's introduction, the author explains why the reader should be interested in his or her research.

What the reviewer said “Authors should summarize the novelty of their research.” was presented in the conclusion section.

Point 9: In Discussion, (page 9 line 247-266) this paragraph is limited to the description of other studies.

Response 9:

The parts mentioned in the review are as follows.

“When each parameter was extracted from three areas, a total of 57 GLCM parameters were extracted. Finally, a total of 58 parameters were used by adding information on the size of the kidney, which is important for the diagnosis of chronic kidney disease. A total of 58 input parameters were tested by constructing an ANN, which is a machine learning method. Input parameters were set to 58, and the hidden layer was set to 10. Since 10 or more hidden layers showed no effect on the classification rate, the experiment was conducted with 10 hidden layers. The three outputs to be classified were normal, early CKD, and end-stage CKD. It is thought that the classification rate of the result to be classified was increased by using all 58 input parameters. The classification accuracy was 95.4%, the epoch needed for training was 38 times, and the misclassification rate was 4.6%. In this experiment, three types of conditions were classified, namely, normal, early CKD, and end-stage CKD, but further detailed classification of disease states is required. In addition, the experiment was conducted by acquiring 741 data items, which is due to the fact that the amount of data is not large; thus, the machine learning method was selected. In the future, a larger amount of data will be acquired, and the deep learning method will be applied to the experiment. “

  1. Conclusion

Chronic kidney disease can be treated if it is detected early, but as the disease progresses, recovery becomes impossible. Eventually, renal replacement therapy such as transplantation or dialysis must be used. In other words, it is crucial to detect and treat chronic kidney disease in the early stages.”

This is the conclusion of this paper and there is no the description of other studies.

Point 10: Authors should compare their results with others, putting in evidence the differences and the value of their data. It should be re-edited

Response 10:

There are no papers that match the method exactly with this paper, so the comparison with similar papers is presented in the "Disscussion 222-234 line".

If you need a format or addition to the method of comparing it with other papers, please point out at the next revision and we will correct it.

Point 11: I suggest English editing to improve the readability of the manuscript.

Response 11:

This paper has received MDPI English service.

We have requested service again to improve our English.

Please wait for the English revised version to arrive.

We will fix it as soon as the revision comes.

Reviewer 2 Report

The title of the article is "Classification of Chronic Kidney Disease in Sonography using 2 the GLCM and artificial neural network". Their efforts for this work were actually huge. The overall writing style, study methodologies and statistical analysis are good. The study was carefully done and reported. I think that this manuscript is interesting. However, there are some flaws, which should be resolved.

  1. The definition of CKD from KDIGO guideline in part of imaging defined “Imaging techniques allow the diagnosis of diseases of the renal structure, vessels and/or collecting systems. Thus, patients with significant structural abnormalities are considered to have CKD if the abnormality persists for greater than 3 months (note that this does not include simple cysts and clinical context is required for action). The diseases are included polycystic kidneys, dysplastic kidneys, hydronephrosis due to obstruction, cortical scarring due to infarcts, pyelonephritis or associated with vesicoureteral reflux, renal masses or enlarged kidneys due to infiltrative diseases, renal artery stenosis, small and hyperechoic kidneys (common in more severe CKD due to many parenchymal diseases)” Although, the authors state that cases with kidney cancer, renal cysts, and congenital diseases of the kidney were excluded from the experiment. How the authors deal with the cases of hydronephrosis due to obstruction, cortical scarring due to infarcts, renal artery stenosis, and single kidney?
  2. How can this technique improve accuracy of diagnoses CKD comparing with previous report? Please clarify, explicit.

Reference: Ma F, Sun T, Liu L, Jing H. Detection and diagnosis of chronic kidney disease using deep learning-based heterogeneous modified artificial neural network. Future Generation Computer Systems. 2020 Oct 1;111:17-26.

Author Response

Response to Reviewer 2 Comments

Thank you for your delicate and caring dissertation review.

I will do the best I can to answer.

Point 1: The definition of CKD from KDIGO guideline in part of imaging defined “Imaging techniques allow the diagnosis of diseases of the renal structure, vessels and/or collecting systems. Thus, patients with significant structural abnormalities are considered to have CKD if the abnormality persists for greater than 3 months (note that this does not include simple cysts and clinical context is required for action). The diseases are included polycystic kidneys, dysplastic kidneys, hydronephrosis due to obstruction, cortical scarring due to infarcts, pyelonephritis or associated with vesicoureteral reflux, renal masses or enlarged kidneys due to infiltrative diseases, renal artery stenosis, small and hyperechoic kidneys (common in more severe CKD due to many parenchymal diseases)” Although, the authors state that cases with kidney cancer, renal cysts, and congenital diseases of the kidney were excluded from the experiment. How the authors deal with the cases of hydronephrosis due to obstruction, cortical scarring due to infarcts, renal artery stenosis, and single kidney?

Response 1:

All patients with diseases mentioned (hydronephrosis due to obstruction, cortical scarring due to infarcts, renal artery stenosis, and single kidney) in the review were excluded.

5 Stages of Kidney Disease

Kidney Function/GFR

Description

Stage 1

> 90%

Normal or High Function

Stage 2

60-89%

Mildly Decreased Function

Stage 3

30-59%

Mild to Moderately

Decreased Function

Stage 4

15-29%

Severely Decreased

Function

Stage 5

<15%

Kidney Failure

It was classified only by the change in the GFR value shown in the table above.

Excluding serious diseases, patients who can only be classified by GFR value were targeted.

Point 2: How can this technique improve accuracy of diagnoses CKD comparing with previous report? Please clarify, explicit.

Reference: Ma F, Sun T, Liu L, Jing H. Detection and diagnosis of chronic kidney disease using deep learning-based heterogeneous modified artificial neural network. Future Generation Computer Systems. 2020 Oct 1;111:17-26

Response 2:

The paper above is a paper comparing the types of artificial neural networks.

It was suggested that HMANN presented in this paper has a higher classification rate than other methods.

This paper is not a paper to develop artificial neural network modeling.

The classification rate is different depending on what the input value is to the existing artificial neural network model. This paper classified CKD using the kidney size and texture analysis parameters used to diagnose CKD in clinical practice.

It is difficult to accurately compare the presented thesis and the research topic of this thesis.

Reviewer 3 Report

*Kim and Ye presented an interesting study: Classification of Chronic kidney disease in sonography using the GLCM and artificial neural network. * *The authors addressed an interested topic, enrolled 741 images, used an interesting methodology, highlighted the strengths of their study, and presented some interesting results and conclusions.* *Some minor comments: * *1.**The authors should describe more precisely to which output layers the misclassification rate is addressed.*** *2.**The authors should clarify: if said end-stage CKD does this correspond to dialysis dependent or transplant kidney status?*** *3.**Does status of early CKD correspond to GFR stage 1 and 2?*** *4.**What was the duration time needed for procedure in total?***

Author Response

Response to Reviewer 3 Comments

Thank you for your delicate and caring dissertation review.

I will do the best I can to answer.

*Kim and Ye presented an interesting study: Classification of Chronic kidney disease in sonography using the GLCM and artificial neural network. * *The authors addressed an interested topic, enrolled 741 images, used an interesting methodology, highlighted the strengths of their study, and presented some interesting results and conclusions.* *Some minor comments:

Point 1: * *1.**The authors should describe more precisely to which output layers the misclassification rate is addressed.

Response 2:

It is not possible to determine exactly which images have been properly classified. Therefore, many studies work on how to classify images in all cases well. It should be used as a diagnostic aid because it is difficult to classify 100% in the part that is automatically diagnosed using a computer.

Point 2: *** *2.**The authors should clarify: if said end-stage CKD does this correspond to dialysis dependent or transplant kidney status?

Response 2:

5 Stages of Kidney Disease

Kidney Function/GFR

Description

Stage 1

> 90%

Normal or High Function

Stage 2

60-89%

Mildly Decreased Function

Stage 3

30-59%

Mild to Moderately

Decreased Function

Stage 4

15-29%

Severely Decreased

Function

Stage 5

<15%

Kidney Failure

The word of the stage was changed by another reviewer. The end-stage you mentioned has been changed to severe. Severe refers to the stage just before dialysis. Based on the criteria presented in the table, steps 2 and 3 were set as mild&moderate, and step 4 was set as severe.

Point 3: *** *3.**Does status of early CKD correspond to GFR stage 1 and 2?***

Response 3:

Yes .

CKD has been classified based on the table presented above.

Point 4:*4.**What was the duration time needed for procedure in total?***

Response 4:

         There are 741 data analyzed, and this process takes less than 10 seconds.

Currently, there is not much data, so the time it takes to perform is meaningless. Because it performs quickly. When the amount of data increases in the future, it is necessary to explore various research methods in consideration of the time performed.

Reviewer 4 Report

-paper should be interesting;;; 
-please add a colorful image of measurements (optionally, if any);;; + arrows what is what;;;;
-please add a block diagram of the proposed method/research ;;;;
-please add photos of the application of the proposed research ;;; 2,3 photos
-please add sentences about future analysis;;;
-Figures should have better quality;;;;
-please add arrows to photos what is what;;; 
-formulas and fonts should be formatted;;;;
-fonts in figures should be bigger;;;;
-labels to figures should be added + SI units if any,
-references should be 2019-2021 Web of Science about 50% or more;; 30-40 at least.;;;;
-what is the accuracy of the proposed approach??? ;;;;
-is there a possibility to use the proposed methods for other problems/compare?  ;;;;

1)
Irfan, M.; Iftikhar, M.A.;
Yasin, S.; Draz, U.; Ali, T.; Hussain, S.;
Bukhari, S.; Alwadie, A.S.;
Rahman, S.; Glowacz, A.; et al. Role
of Hybrid Deep Neural Networks
(HDNNs), Computed Tomography,
and Chest X-rays for the Detection of
COVID-19. Int. J. Environ. Res. Public
Health 2021, 18, 3056. 
https://doi.org/10.3390/ijerph18063056

maybe
2)
Szaleniec J, Szaleniec M, Strek P, Boron A, Jablonska K, Gawlik J, SkÅ‚adzien J. Outcome prediction in endoscopic surgery for chronic 
rhinosinusitis - a multidimensional model. Adv Med Sci. 2014 Mar;59(1):13-8. doi: 10.1016/j.advms.2013.06.003. Epub 2014

-Conclusion: point out what are you done;;;;

Author Response

Response to Reviewer 4 Comments

Thank you for your delicate and caring dissertation review.

I will do the best I can to answer.

Point 1: paper should be interesting;;; 

please add a colorful image of measurements (optionally, if any);;; + arrows what is what;;;;

Response 1:

Ultrasound images are originally black and white images. An explanation was added using arrows in the ultrasound image.

Figure 1 shows three categories of images. The reason for including these images to show the reader about the normal, mild and moderate and Severe CKD.

Figure 4 added a description using arrows.

Point 2: please add a block diagram of the proposed method/research ;;;;

Response 2:

Figure 2 is a block diagram of the proposed methods.

Please tell us what to add to the picture 2 content and we will change it.

Point 3: please add photos of the application of the proposed research ;;; 2,3 photos

Response 3:

   Figure 3 shows the application program implementation results.

Point 4: please add sentences about future analysis.

Response 4:

Modified and added content for future analysis.

Point 5: Figures should have better quality

Response 5:

The Figures are added in high quality.

In the process of putting the image in the body, I had to put the picture in small size to fit
the space. Whether it is a word file or a pdf file, you can see the contents sufficiently because the image quality is high when you enlarge it.

Figure 6 is an ROI image with a resolution of 50*50. So, the image shown is the best.

Point 6: please add arrows to photos what is what;;; 

Response 6:

Figure 4 added a description using arrows.

Point 7: formulas and fonts should be formatted;;;;

Response 7:

     The font in the table has been modified according to the style.

Point 8: fonts in figures should be bigger

Response 8:

At the Figure 2, Figure 4 , Increased the font size.

Point 9: labels to figures should be added + SI units if any,

   Response 9:

     When applying GLCM to an image, each parameter of GLCM is obtained using the value of each pixel in the image. At this time, the unit of each parameter cannot be expressed in SI units. Many papers do not indicate SI units to interpret images using values ​​for each parameter.

Point 10: references should be 2019-2021 Web of Science about 50% or more;; 30-40 at least.;;;;

Response 10:

     More than 50% of the latest(2019-2021) papers have been revised.

Point 11: what is the accuracy of the proposed approach??? ;;;;

   Response 11:

   The accuracy of the results is indicated by the ROC curve.(Figure 8)

Point 12: is there a possibility to use the proposed methods for other problems/compare?  

  1. Irfan, M.; Iftikhar, M.A.;Yasin, S.; Draz, U.; Ali, T.; Hussain, S.;Bukhari, S.; Alwadie, A.S.;Rahman, S.; Glowacz, A.; et al. Roleof Hybrid Deep Neural Networks(HDNNs), Computed Tomography,and Chest X-rays for the Detection ofCOVID-19. Int. J. Environ. Res. PublicHealth 2021, 18, 3056. https://doi.org/10.3390/ijerph18063056
  2. Szaleniec J, Szaleniec M, Strek P, Boron A, Jablonska K, Gawlik J, SkÅ‚adzien J. Outcome prediction in endoscopic surgery for chronic  rhinosinusitis - a multidimensional model. Adv Med Sci. 2014 Mar;59(1):13-8. doi: 10.1016/j.advms.2013.06.003. Epub 2014 Conclusion: point out what are you done;;;;
  3.  

Response 12: The GLCM algorithm proposed in this paper is likely to be used mainly for ultrasound imaging. This is because it exhibits good texture properties. Our study was mainly used for ultrasound imaging. For example, it was used for ultrasound imaging of the breast, thyroid gland, and fetus. In artificial neural networks, the input value acts as an important factor for classification. The artificial neural network used an existing program.So what is the input node is important.In the first document, the possibility of application to X-ray images is considered sufficient.It is thought that it is necessary to find a method other than GLCM for input parameters.In the second paper, multidimensional models (artificialNeural networks) are poorly explained, making it difficult to understand modeling. In other words, it seems that the reference to the input variable to the model is ambiguous. If the definition of modeling is clear, the result is similar to that of our paper. However, when configuring modeling, how to configure the input and hidden layers is a very important issue. It is difficult to compare papers without these references to our papers.

Round 2

Reviewer 1 Report

The authors have addressed all the issues raised

Reviewer 2 Report

The authors had adequate addressed my previous comments. I have no additional concerns.

Reviewer 4 Report

-photo of application of the proposed system can be added